# The Last Mile to Supervised Performance: Semi-Supervised Domain Adaptation for Semantic Segmentation

## Abstract

Supervised deep learning requires massive labeled datasets, but obtaining annotations is not always easy or possible, especially for dense tasks like semantic segmentation. To overcome this issue, numerous works explore Unsupervised Domain Adaptation (UDA), which uses a labeled dataset from another domain (source), or Semi-Supervised Learning (SSL), which trains on a partially labeled set. Despite the success of UDA and SSL, reaching supervised performance at a low annotation cost remains a notoriously elusive goal. To address this, we study the promising setting of Semi-Supervised Domain Adaptation (SSDA). We propose a simple SSDA framework that combines consistency regularization, pixel contrastive learning, and self-training to effectively utilize a few target-domain labels. Our method outperforms prior art in the popular GTA→Cityscapes benchmark and shows that as little as 50 target labels can suffice to achieve near-supervised performance. Additional results on Synthia→Cityscapes, GTA→BDD and Synthia→BDD further demonstrate the effectiveness and practical utility of the method. Lastly, we find that existing UDA and SSL methods are not well-suited for the SSDA setting and discuss design patterns to adapt them.

## 1 Introduction

Semantic segmentation is a key task in computer vision with diverse applications ranging from autonomous driving (Badrinarayanan et al., 2017) to medical image analysis (Ronneberger et al., 2015). Despite recent progress in this area using supervised learning methods (Badrinarayanan et al., 2017; Ronneberger et al., 2015; Xie et al., 2021), supervision remains challenging in practical applications due to the high labeling cost and the need for specialized domain experts. Therefore, minimizing the labeling cost while maintaining strong performance is critical. Common approaches for learning with unlabeled data are Unsupervised Domain Adaptation (UDA), which uses additional data from another similar domain, and Semi-Supervised Learning (SSL), which trains on a partially labeled set.

While UDA has demonstrated promising results on public benchmarks, its practical implementation remains challenging. Although UDA methods do not require target annotations for training and leverage additional labeled data from a source domain, they often require target labels for hyperparameter tuning (Saito et al., 2021). Moreover, it is essential in industrial and medical applications to have a well-validated system, which necessitates the collection of a target labeled set for validation purposes. In such cases, annotating a few samples for training may not a significant overhead. Another setting to learn with missing labels is SSL, which trains a model on a partially labeled dataset (Chen et al., 2021b; Alonso et al., 2021; Olsson et al., 2021). However, SSL methods may underperform and risk overfitting when the number of labels is low. While adding a source dataset can alleviate this problem, existing SSL methods are not designed to leverage data from another domain, and studies like the one of Alonso et al. (2021) have shown only moderate improvement. Despite the competitive performance of both UDA and SSL methods, they fall short of supervised performance, as they achieve significantly lower accuracies than the fully supervised counterpart.

In this work, we study how to close the gap to supervised performance by exploiting the Semi-Supervised Domain Adaptation (SSDA) setting, and show that it is possible to match supervised accuracy at a modest

Table 1: **Summary of settings.** Types of data used in Semi-Supervised Learning (SSL), Unsupervised Domain Adaptation (UDA) and Semi-Supervised Domain Adaptation (SSDA).

| Data | Source Labeled | Target Labeled | Target Unlabeled |
|---|---|---|---|
| SSL | ✗ | ✓ | ✓ |
| UDA | ✓ | ✗ | ✓ |
| SSDA | ✓ | ✓ | ✓ |

annotation cost. SSDA is essentially the combination of SSL and UDA, as it uses source labeled data, target unlabeled data, and a few target labels (Tab. 1). Despite its practical value and performance potential while alleviating annotation requirements, SSDA has received less attention (Berthelot et al., 2021). To our knowledge, only two works present a semantic segmentation method tailored to SSDA (Wang et al., 2020b; Chen et al., 2021a), and Alonso et al. (2021) propose an SSL method and try to extend it to SSDA. Moreover, the existing UDA works do not explore incorporating a few target labels and are suboptimal in an SSDA setting.

We introduce a simple and straightforward semantic segmentation framework tailored to SSDA, which uses a combination of consistency regularization (CR) and pixel contrastive learning (PCL). The main goal of the method is to achieve compact clusters of target representations, which facilitate the classification task, while also learning a domain-robust feature extractor to leverage the source domain data. Moreover, we also focus on effectively utilizing the few available target labels. Finally, we propose a self-training scheme that improves training efficiency by iteratively refining model and pseudolabels. Our comprehensive evaluation on GTA→Cityscapes demonstrates how the proposed method achieves state-of-the-art performance on SSDA semantic segmentation, approaching the supervised performance with minimal annotation, using $\leq 1/15$ of target labels (see Fig. 1). Additional results in other benchmarks, without further hyperparameter tuning, confirm the effectiveness and high practical value of the method. We will make the code available upon acceptance.

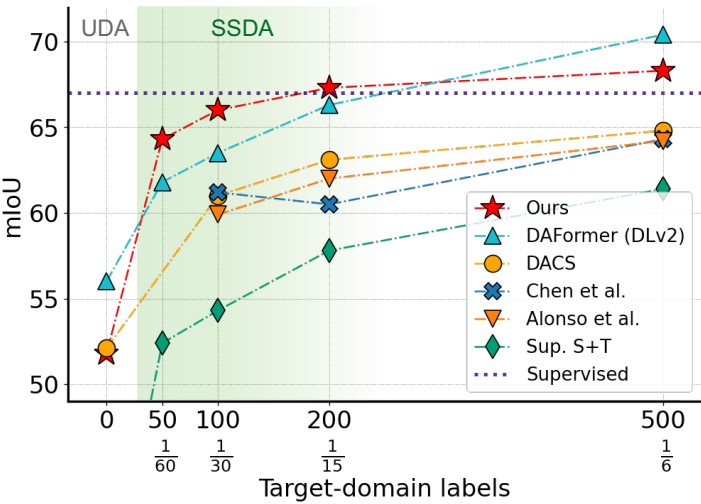

Figure 1: GTA→Cityscapes results (mIoU). Our method beats all baselines in the highlighted regime of interest: SSDA with a low amount of target labels. We claim SSDA as an alternative to UDA where near-supervised performance can be achieved at a low annotation cost. "Supervised" indicates a model trained on the full target dataset (2975 images). Fractions represent ratio of target-domain samples labeled. Results are an average of 3 runs on a DeepLabv2 + ResNet-101 network. See Tab. 2 for the results table.

The main contributions of this paper are:

- We present a simple SSDA method for semantic segmentation that effectively utilizes the different kinds of data available, reaching a performance comparable to supervised learning.

- We demonstrate a significant improvement of SSDA over UDA even with only 50 target labels (+6.9 mIoU). We also find that existing UDA methods are suboptimal in SSDA and discuss potential avenues for adapting them.

- We investigate the relationship between SSL and SSDA, and show an improvement over the former (+9.0 mIoU at 50 labels) when effectively leveraging source domain data.

## 2 Related Work

### 2.1 Unsupervised Domain Adaptation (UDA) for semantic segmentation

Numerous approaches have been proposed for UDA in semantic segmentation. In recent years, these techniques have been broadly classified into two main categories: adversarial training and self-training. Adversarial training methods minimize the difference between the source and target domains through a minimax game between a feature extractor and a domain discriminator (Ganin et al., 2016; Hoffman et al., 2018; Vu et al., 2019; Wang et al., 2020a). Conversely, self-training methods involve producing pseudolabels for the target domain data and aligning the two domains by means of domain mixing (Tranheden et al., 2021) or source styling (Yang & Soatto, 2020). Pseudolabels can be carefully generated using prototypes (Zhang et al., 2021; Liu et al., 2021) or adaptive confidence thresholds (Mei et al., 2020). An iterative self-training algorithm that employs pseudolabels is explored by Zou et al. (2018) and Li et al. (2019). While all the above-mentioned methods employ the Deeplab family of architectures (Chen et al., 2017b; 2018), recent studies have shown that self-training methods using Transformer-based networks have achieved state-of-the-art performance (Hoyer et al., 2021a; 2022). Lastly, Hoyer et al. (2022) utilize high resolution and multi-scale inputs with a module that can be applied on top of existing UDA methods. Even though several ideas from UDA can potentially be employed in SSDA frameworks, out-of-the-box UDA methods are suboptimal in SSDA (see 4.2.3), since they do not consider how to fully leverage the few, very valuable, target labels. Therefore, to fully leverage the provided labels, we need to design frameworks tailored to SSDA.

### 2.2 Semi-Supervised Learning (SSL) for semantic segmentation

Learning on a partially labeled dataset has been largely explored for semantic segmentation. A commonly used mechanism is consistency regularization, which aims to learn a model invariant to perturbations by encouraging consistent predictions between augmentations of an unlabeled image. Relying on the cluster and smoothness assumptions (Chapelle & Zien, 2005), it encourages compact clusters of representations separated by low-density regions, where the decision boundary can lie. Some approaches use a mean teacher to generate pseudolabels (Tarvainen & Valpola, 2017; French et al., 2019; Alonso et al., 2021; Liu et al., 2022c), while others train a single model (Sohn et al., 2020; Zou et al., 2020) or perform cross-supervision between two models (Chen et al., 2021b; Fan et al., 2022; Ke et al., 2020).

Iterative self-training consists of training for one round and using the resulting model to generate pseudolabels to train a new model in the next round (Xie et al., 2020; Zoph et al., 2020; Zou et al., 2020; Teh et al., 2022; Liu et al., 2022a). In contrast to consistency regularization, the pseudolabels are generated offline. Despite its effectiveness, self-training can suffer from using noisy pseudolabels or perpetuate a model bias. Unsupervised pixel contrastive learning has been used in SSL to encourage compact clusters of representations (Alonso et al., 2021; Kwon & Kwak, 2022; Liu et al., 2022b). This mechanism pulls together positive pairs of pixels in the latent space, while pulling negative pairs apart to increase separability. Moreover, supervised pixel contrastive learning has been proposed as a regularizer of the embedding space to encourage better clusterability (Wang et al., 2021; Pissas et al., 2022), boosting the performance of fully supervised methods. Even if SSL frameworks may share some elements with SSDA methods, they should be properly modified to account for domain adaptation in order to leverage source domain data. The interplay between DA and SSL mechanisms, which we study in this work, is not trivial to predict and requires careful consideration.

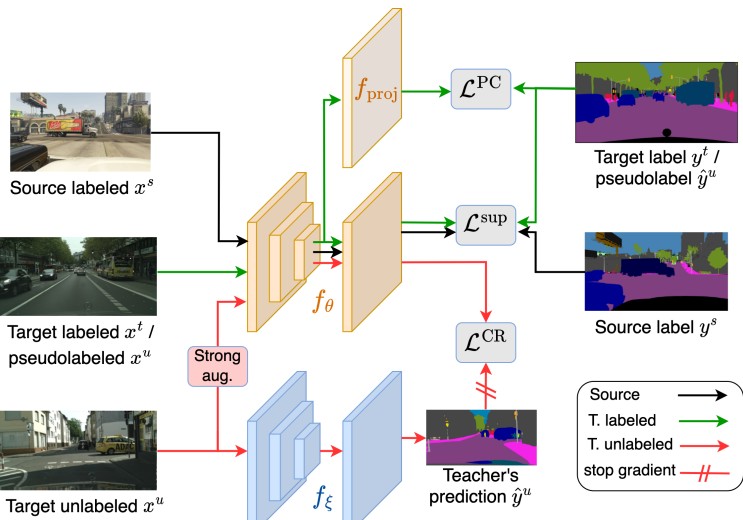

Figure 2: **Framework overview.** In each round, we train a student model $f_\theta$ with a combination of supervised learning $\mathcal{L}^{\text{sup}}$, consistency regularization (CR) $\mathcal{L}^{\text{CR}}$ and pixel contrastive learning $\mathcal{L}^{\text{PC}}$. We use a mean teacher $f_\xi$ to generate pseudotargets in CR, and stop its gradient. In subsequent rounds of self-training, the target labeled set includes pseudolabels generated in the previous round.

## 2.3 Semi-Supervised Domain Adaptation (SSDA)

In SSDA the learner has access to source labels, target unlabeled data and a few target labels. SSDA is less explored in the literature, only recently it has received more attention in image classification (Saito et al., 2019; Berthelot et al., 2021; Qin et al., 2021; Kim & Kim, 2020). While most methods are based on UDA's core idea of domain alignment, (Mishra et al., 2021) notice that a few target labels are sufficient in SSDA to forego domain alignment and focus on target feature clusterability instead. However, the dense task of semantic segmentation is more complex than image classification, requiring SSDA methods to be revisited and developed for this setting. Additional challenges of the task are the uncertainty in pixels (e.g., at boundaries between objects), which impedes the use of explicit entropy minimization (Saito et al., 2019), and a large class imbalance.

So far, two frameworks have been devised for SSDA in semantic segmentation (Wang et al., 2020b; Chen et al., 2021a), and one more considers the extension from SSL (Alonso et al., 2021). Wang et al. (2020b) uses adversarial training to align the domains at two representation levels, local and global, but fails to fully leverage the few target labels. Chen et al. (2021a) base their method on domain mixing and iterative self-training, with the goal of aligning source and target domain representations. The domain mixup is achieved with CutMix (Yun et al., 2019) and by mixing domains in the mini-batch. Lastly, Alonso et al. (2021) propose an SSL framework with consistency regularization and pixel contrastive learning. They also investigate the extension to SSDA by adding source data, but only find a moderate improvement since they do not take domain alignment considerations.

## 3 Method

In this section, we present our framework for SSDA semantic segmentation. In SSDA, we have access to a source labeled dataset $\mathcal{D}_s = \{(x_i^s, y_i^s)\}_{i=1}^{N_s}$, a few target labeled samples $\mathcal{D}_t = \{(x_i^t, y_i^t)\}_{i=1}^{N_t}$ and a set of target unlabeled samples $\mathcal{D}_u = \{x_i^u\}_{i=1}^{N_u}$, where typically $N_t \ll N_u$.

The main goal of our framework is to encourage tight clustering of target representations, such that similar pixels are clustered together in the latent space and the identity of each cluster is inferred from the few labels, a key idea in SSL. Moreover, we consider domain alignment to better leverage source data, such that

source and target representations are aligned and the model can generalize to both domains. A schematic of the framework is depicted in Fig. 2. We use a student-teacher scheme, keeping a set of parameters $\theta$ for the student model $f_\theta$ and parameters $\xi$ for the teacher model $f_\xi$. The teacher model $f_\xi$ is an exponential moving average (EMA) of $f_\theta$ with coefficient $\mu \in [0,1]$, which provides more robust predictions (Tarvainen & Valpola, 2017). The parameters of $f_\xi$ are updated by $\xi = \mu\xi + (1-\mu)\theta$.

In the next subsections we present each of the components of the framework: a supervised objective (Sec. 3.1), consistency regularization (Sec. 3.2), pixel contrastive learning (Sec. 3.3) and an iterative self-training scheme (Sec. 3.4). Finally, in Sec. 3.5 we discuss how to extend the framework to the neighboring settings of UDA and SSL.

## 3.1 Supervised training on labeled data

The available source and target labels are used in a supervised fashion to minimize the cross-entropy with respect to the model predictions. We use class weights to mitigate the class imbalance in semantic segmentation datasets. Importantly, we mix source and target batches which helps in learning domain-robust representations (Chen et al., 2021a). We define $\mathcal{L}^{\mathrm{sup}}$ as

$$\mathcal{L}^{\mathrm{sup}} = \lambda_s \, \mathcal{Q}(f_\theta(x^s), y^s) + \lambda_t \, \mathcal{Q}(f_\theta(x^t), y^t), \tag{1}$$

where $\mathcal{Q}(\cdot, \cdot)$ is the weighted cross-entropy. With images of $H \times W$ pixels, one-hot semantic labels as $y$ and $C$ classes, $\mathcal{Q}(\cdot, \cdot)$ is defined as

$$\mathcal{Q}(\hat{y}, y) = -\frac{1}{H \cdot W} \sum_{j=1}^{H \cdot W} \sum_{c=1}^{C} \alpha_c \cdot y_{j,c} \cdot \log(\hat{y}_{j,c}). \tag{2}$$

Class weights $\alpha_c$ are computed for $\mathcal{D}_s$ and $\mathcal{D}_t$ separately (see Sec. 8).

## 3.2 Consistency Regularization

Consistency regularization is an unsupervised mechanism that encourages tight and well-separated clusters of representations by promoting consistent predictions between different augmentations of an image. We define $\mathcal{L}^{\mathrm{CR}}$ as the pixel-wise cross-entropy between the prediction of the student $f_\theta$ on a random strong augmentation $x'$ and a one-hot pseudo-target generated by the teacher $f_\xi$ on the original image $x$. The gradient is stopped on the pseudo-target such that $f_\xi$ does not receive any update. The consistency loss for an image $x$ is

$$\hat{y}_j = \arg\max f_\xi(x)_j, \tag{3}$$

$$\mathcal{L}^{\mathrm{CR}}(x) = \frac{1}{H \cdot W} \sum_{j=1}^{H \cdot W} \mathrm{CE}\big(f_\theta(x')_j, \hat{y}_j\big), \tag{4}$$

where $\mathrm{CE}(\cdot, \cdot)$ is the standard cross-entropy loss. This objective leverages unlabeled target data $\mathcal{D}_u$. Details on the transformations used for the random augmentations are provided in Sec. C.

## 3.3 Supervised Pixel Contrastive Learning

To further enhance target feature clusterability, we add a pixel contrastive objective for target labeled data $\mathcal{D}_t$. With this objecive, pixels of the same class are pushed together in the embedding space, forming more compact clusters, while pixels of different classes are pushed apart, forming low-density regions between clusters. A projection head $f_{\mathrm{proj}}$ produces pixel embeddings $z_j$ to be contrasted. The supervised contrastive loss for pixel $j$ is given by

$$\mathcal{L}_j^{\mathrm{PC}} = \frac{1}{|\mathcal{P}_j|} \sum_{z_j^+ \in \mathcal{P}_j} -\log \frac{\exp(z_j \cdot z_j^+/t)}{\exp(z_j \cdot z_j^+/t) + \sum_{z_j^- \in \mathcal{N}_j} \exp(z_j \cdot z_j^-/t)}, \tag{5}$$

where $z_j$ is contrasted with a set $\mathcal{P}_j$ of positive samples from the same class and a set $\mathcal{N}_j$ of negative samples from different classes. The symbol $t$ denotes a temperature hyperparameter. A more complex version of this module was introduced by Wang et al. (2021), but we found the memory bank or the pixel-to-region contrast redundant in our preliminary experiments. Importantly, we apply this objective to target labeled data only, and not unlabeled samples, as relying on ground-truth results in better learnt representations. At each iteration we contrast a subset of pixels sampled from the current batch, up to $N_{\text{pix}}$ from each class, using hard example sampling (Wang et al., 2021). Let $A$ be the total number of pixels sampled from the $\mathcal{D}_t$ batch, with $A \leq N_{\text{pix}} \cdot C$, then the pixel contrastive loss is defined by

$$\mathcal{L}^{\text{PC}} = \frac{1}{A} \sum_{j=1}^{A} \mathcal{L}_j^{\text{PC}}. \tag{6}$$

Collecting equation 1, equation 4 and equation 6, the overall loss function to be minimized is given by

$$\mathcal{L} = \mathcal{L}_{\mathcal{D}_s, \mathcal{D}_t}^{\text{sup}} + \lambda_1 \, \mathcal{L}_{\mathcal{D}_u}^{\text{CR}} + \lambda_2 \, \mathcal{L}_{\mathcal{D}_t}^{\text{PC}}. \tag{7}$$

We minimize $\mathcal{L}$ in each iteration of a self-training scheme, explained in the next section.

### 3.4 Iterative Self-training

In the few-labels regime, the lack of diversity in $\mathcal{D}_t$ is problematic. To mitigate that we employ an offline self-training algorithm that leverages pseudolabels for unlabeled images in $\mathcal{D}_u$. A more diverse pool of labeled samples increases the efficiency of training. In a second stage of each iteration, we drop the pseudolabels, which innevitably contain some noise, to fine-tune using only ground-truth annotations. The procedure is summarized in Algorithm 1, where $\mathbf{M}_k$ represents a model trained in the $k^{\text{th}}$ self-training round. The quality of psuedolabels is critical in self-training. Following Li et al. (2019), we only annotate pixels with a prediction confidence above a threshold $\tau$, and discard the pseudolabel on pixels with uncertain predictions.

---

**Algorithm 1** Iterative Self-training

Train $\mathbf{M}_0$ on $\left[\mathcal{D}_s, \mathcal{D}_t, \mathcal{D}_u\right]$ for $n_{\text{steps}}$.         ▷ First training round

**for** $k = \{1, \ldots, K\}$ **do**         ▷ Self-training rounds
    $\{\hat{y}_i^u\}_{i=1}^{N_u} = \texttt{generate\_PL}(\mathcal{D}_u, \mathbf{M}_{k-1})$
    $\mathcal{D}_{t+\hat{u}} \leftarrow \mathcal{D}_t \cup \{(x_i^u, \hat{y}_i^u)\}_{i=1}^{N_u}$
    Train $\mathbf{M}_k$ on $\left[\mathcal{D}_s, \mathcal{D}_{t+\hat{u}}, \mathcal{D}_u\right]$ for steps $[0, n_{\text{drop}})$
    Train $\mathbf{M}_k$ on $\left[\mathcal{D}_s, \mathcal{D}_t, \mathcal{D}_u\right]$ for steps $[n_{\text{drop}}, n_{\text{steps}})$
**end for**

Return $(\mathbf{M}_{K-1}, \mathbf{M}_K)$.         ▷ Use ensemble at test time

---

### 3.5 Adaptation to UDA and SSL

In this section we discuss how to adapt our SSDA framework to be used in the UDA and SSL settings. For SSL we simply drop the source data and the supervised loss term becomes $\mathcal{L}^{\text{sup}} = \lambda_t \, \mathcal{Q}(f_\theta(x^t), y^t)$. The adaption to UDA has two caveats. Firstly, since we do not have target labeled data, we cannot apply the pixel contrastive learning module on $\mathcal{D}_t$. Therefore, we only use this module on $\mathcal{D}_{t+\hat{u}}$ when pseudolabels are available. Secondly, we modify the consistency regularization formulation to use the teacher's class probability predictions as pseudo-targets, instead of transforming them into a one-hot encoding. Thus, equation 4 is replaced by

$$\mathcal{L}_{\text{prob}}^{\text{CR}}(x) = \frac{1}{H \cdot W} \sum_{j=1}^{H \cdot W} \text{CE}\big(f_\theta(x')_j, \, f_\xi(x)_j\big). \tag{8}$$

We observed that using $\mathcal{L}_{\text{prob}}^{\text{CR}}$ resulted in more stable training in UDA, while $\mathcal{L}^{\text{CR}}$ was stable in SSDA and yielded a slighlty better performance.

# 4 Experiments

This section presents the experimental setup, SSDA results of the proposed framework, a comparison to UDA and SSL methods, and ablation studies.

## 4.1 Implementation Details

Below we discuss the datasets and model architecture used. As for hyperparamters, we use a fixed training configuration across all experiments and for all datasets, which is detailed in Tab. 8 in the Appendix. Experiments are conducted on a single V100 GPU with 32 GB of memory.

### 4.1.1 Datasets

We use the popular GTA→Cityscapes as our main semantic segmentation benchmark. Cityscapes, the target dataset, has 2975 training and 500 validation images of European urban scenarios, manually annotated with 19 classes. As standard, we downsample the original resolution of $2048 \times 1024$ pixels to $1024 \times 512$ for training. The source GTA dataset (Richter et al., 2016) contains 24966 computer-generated urban images for training, which we downsample from $1914 \times 1052$ to $1280 \times 720$ pixels, as standard. Labels contain 33 semantic classes, we select only the 19 classes that coincide with Cityscapes, as Wang et al. (2020b).

Additionally, we experiment on the datasets of Synthia (source) and BDD (target). Synthia (Ros et al., 2016) has 9400 synthetic images of $1280 \times 760$ pixels. It is evaluated on 16 or 13 classes, also present in Cityscapes. For BDD (Yu et al., 2020) we use the 7000 train and 1000 validation real images of US streets at the original resolution of $1280 \times 720$ pixels.

For all datasets, we perform random square crops of $512 \times 512$ and horizontal flips at training time. For evaluation, following standard procedure, we report the mean Intersection over Union (mIoU), averaged over 3 runs with different random labeled/unlabeled training set split.

### 4.1.2 Architecture

We use a DeepLabv2 (Chen et al., 2017a) decoder and ResNet-101 backbone, for fair comparison with previous works on SSDA (Wang et al., 2020b; Chen et al., 2021a; Alonso et al., 2021), and which is also widely used in UDA benchmarks. The DeepLabv2 decoder uses an ASPP module to obtain multi-scale representations. The ResNet backbone used is always pretrained on ImageNet. Following Wang et al. (2021), the projection head $f_{\text{proj}}$ for pixel contrast transforms the 2048-dim features from the backbone into 256-dim normalized embeddings. It is composed of two $1 \times 1$ convolutional layers interleaved with `ReLU` and `BatchNorm` layers.

## 4.2 Results

### 4.2.1 SSDA on GTA→Cityscapes

We present our main SSDA results on the widely used GTA→Cityscapes benchmark in Tab. 2 and Fig. 1. We compare our performance to the existing SSDA semantic segmentation methods. Moreover, to provide a competitive baseline, we extend a state-of-the-art UDA method (DAFormer, Hoyer et al. (2021a)) to SSDA. We also include results for training only on labeled target (T) or source and target (S+T) data, and a fully supervised (FS) oracle trained on the entire 2975 target labeled samples.

Our framework outperforms all previous methods by a substantial margin and sets a new state-of-the-art in the SSDA regime with few labels ($^1\!/_{60}$, $^1\!/_{30}$ and $^1\!/_{15}$ of labeled data). Furthermore, at the most challenging setting of 50 ($^1\!/_{60}$) target labels, we beat most of the baselines when they use $\times 4$ or even $\times 10$ more labels. Only when labels are more abundant, at 500 ($^1\!/_6$) target labels, does DAFormer outperform ours, which we speculate is due to the specific measures it takes to generalize in Cityscapes, such as thing-class regularization.

Table 2: GTA→Cityscapes SSDA semantic segmentation results (mIoU) with a DeepLabv2 + ResNet-101 network. Our framework outperforms baselines in the SSDA low-label regime, achieving near fully supervised (FS) performance at a low annotation cost. All results are averaged over 3 runs.

| Target labels | UDA | 50 | 100 | 200 | 500 | FS |
|---|---|---|---|---|---|---|
| Label ratio | 0 | $^1/_{60}$ | $^1/_{30}$ | $^1/_{15}$ | $^1/_6$ | 1 |
| $\mathcal{L}^{sup}$ (T) | - | 41.2 | 46.5 | 52.7 | 60.4 | 67.0 |
| $\mathcal{L}^{sup}$ (S+T) | - | 52.4 | 54.3 | 57.8 | 61.4 | 65.8 |
| ASS (Wang et al., 2020b) | - | - | 54.2 | 56.0 | 60.2 | 65.9 |
| Alonso et al. (2021) | - | - | 59.9 | 62.0 | 64.2 | 67.3 |
| DACS (Tranheden et al., 2021) [*] | 52.1 | - | 61.0 | 63.1 | 64.8 | - |
| Chen et al. (2021a) | - | - | 61.2 | 60.5 | 64.3 | 65.3 |
| DAFormer (Hoyer et al., 2021a) [†] | **56.0** | 61.8 | 63.5 | 66.3 | **70.4** | - |
| Ours | 51.8 | **64.3** | **66.0** | **67.3** | 68.3 | 67.0 |

[*] SSDA results from Hoyer et al. (2021b)

[†] on DeepLabv2

Table 3: SSDA semantic segmentation results on GTA→Cityscapes (mIoU) with a DAFormer network (Transformer-based). We extend DAFormer to the SSDA setting and outperform it at the low-label regime, but fall short of supervised (FS) performance. All results are averaged over 3 runs.

| Setting | UDA | SSDA | | | |
|---|---|---|---|---|---|
| Target labels | 0 | 50 | 100 | 200 | 500 |
| Label ratio | 0 | $^1/_{60}$ | $^1/_{30}$ | $^1/_{15}$ | $^1/_6$ |
| *GTA → Cityscapes (DAFormer)* | FS: 77.6 mIoU (Hoyer et al., 2021a) | | | | |
| DAFormer (Hoyer et al., 2021a) | **68.3** | 66.2 | 69.8 | 71.2 | **74.4** |
| Ours | 55.5 | **68.2** | **71.4** | **72.1** | 73.5 |

Compared to supervised performance, with 100 target labels we already achieve an accuracy of 66.0 mIoU, only 1.0 point shy of FS, and surpass it with 200 of labels. Thus, we demonstrate the potential of SSDA to close the gap to supervised performance at a moderate annotation cost.

Our method greatly outperforms the previous works tailored to SSDA segmentation. In particular, we do not find necessary to mix domains explicitly as in Chen et al. (2021a), the implicit mixing by using mixed batches (see Sec. 4.3.1) and the domain robustness effect of consistency regularization (see Sec. 4.3.3) achieve a better domain alignment. We also compare against DAFormer on their Transformer-based architecture (for implementation details see App. E.2). We find that our method outperforms DAFormer in the semi-supervised low-label regime (Tab. 3). However, the gap to supervised performance is still large, interesting future work could be focused on SSDA methods tailored to Transfomers. As Hoyer et al. (2021a) show, this network requires careful design to avoid overfitting to common classes and achieve stable training, which our framework is missing and explains the gap in UDA performance.

### 4.2.2 SSDA on other datasets.

To show the generalization ability of the proposed method to other datasets, we perform experiments on Synthia→Cityscapes, GTA →BDD and Synthia→BDD, all of them *syn2real* semantic segmentation tasks. We focus on the most challenging SSDA regime, with $\leq$ $^1/_{30}$ target labels. We use the same training configuration as for GTA→Cityscapes, without tuning any hyperparameter.

Table 4: SSDA semantic segmentation results (mIoU) on additional benchmarks. Our method achieves near fully supervised (FS) performance at a low annotation cost. All results are averaged over 3 runs on a DeepLabv2 + ResNet-101 network.

| $Synthia{\to}CS$, 16 (13) classes, FS: 68.9 (73.1) mIoU | | | |
|---|---|---|---|
| Target labels | 0 | 50 ($1/60$) | 100 ($1/30$) |
| $\mathcal{L}^{\text{sup}}$ (S+T) | 29.4 (33.6) | 49.0 (58.4) | 52.5 (61.8) |
| Ours | - (-) | **64.5** (**73.9**) | **67.2** (**75.7**) |
| DAFormer (Hoyer et al., 2021a) † | 53.4 (60.6) | 62.4 (68.0) | 64.6 (70.6) |
| $GTA{\to}BDD$, 19 classes, FS: 55.8 mIoU | | | |
| Target labels | 0 | 100 ($1/70$) | 233 ($1/30$) |
| $\mathcal{L}^{\text{sup}}$ (S+T) | 33.2 | 48.3 | 51.5 |
| Ours | 43.1 | **52.5** | **54.5** |
| $Synthia{\to}BDD$, 16 classes, FS: 56.6 mIoU | | | |
| $\mathcal{L}^{\text{sup}}$ (S+T) | 24.2 | 43.5 | 48.1 |
| Ours | - | **54.5** | **57.6** |

† on DeepLabv2.

For all datasets, we show that our method is comparable to or outperforms fully supervised (FS) training using only $1/30$ target labels (Tab. 4). Moreover, for Synthia→Cityscapes we also run experiments on DAFormer to provide a competitive baseline, which we beat in all cases. The positive results in other datasets without changing the hyperparameter configuration suggest high practical applicability of the method proposed.

### 4.2.3 UDA → SSDA.

When no labels are available, our SSDA framework in a UDA setting (see Sec. 3.5) achieves 51.8 mIoU, which is comparable to well-established methods such as DACS (Tranheden et al., 2021), but below recent specialized methods. Compared to UDA state-of-the-art, with BAPA (Liu et al., 2021) achieving an accuracy of 57.4 mIoU, our method improves by +6.9 mIoU using only 50 target labels. This result demonstrates the high value of even just a few annotations and thus the potential of SSDA. In Sec. E.1 we present an extended comparison to UDA methods, including those using high-resolution images (e.g., HRDA (Hoyer et al., 2022)), which we omit here for a fair comparison.

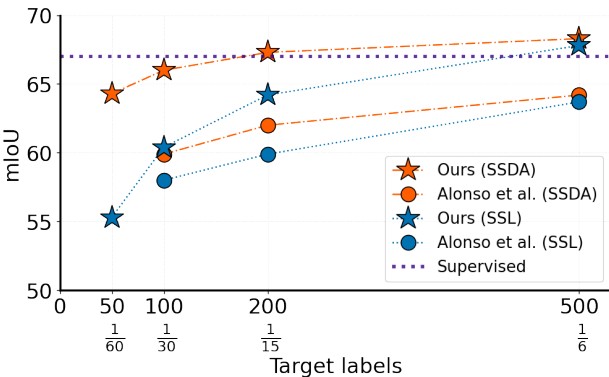

Figure 3: SSL vs. SSDA semantic segmentation results (mIoU) on GTA→Cityscapes for our method and Alonso et al. (2021). We show a substantial improvement when using source data (SSDA) compared to SSL, particularly in the low-label regime. The difference is less pronounced as more target labels are used. All results are the average of 3 runs on a DeepLabv2 with ResNet-101 backbone.

### 4.2.4 SSL → SSDA.

To quantify the potential improvement of using a source domain, in this section compare our SSDA framework to its SSL counterpart. To our knowledge, only Alonso et al. (2021) have compared these settings, which demonstrated only a moderate improvement when using source data. However, we found that when using a framework that takes measures to align domains, adding source domain data can substantially improve performance. In Fig. 3 we show a direct comparison of SSL vs. SSDA between our method and Alonso et al. (2021). Our method better leverages source data and obtains +9.0 mIoU at 50 labels (1/60) and +5.6 mIoU at 100 labels (1/30). Interestingly, we observe a trend where the performance boost of SSDA decreases as more target labels are available, shrinking to +0.5 mIoU at 500 labels. We conclude that a source dataset is particularly beneficial when very few target labels are available, as it reduces the risk of SSL to overfit to the few annotations.

## 4.3 Ablation studies

In this section we explore the impact of each component of the framework.

### 4.3.1 Framework ablation

In Tab. 5 we compare the performance of $\mathbf{M}_0$, a model trained on $\mathcal{L}$ (7) for one training round, to a number of framework variants. We find that consistency regularization is by far the most important element, as removing $\mathcal{L}^{\mathrm{CR}}$ results in $-8$ mIoU. We also find it important to use class weights to mitigate class imbalance ($-2.3$ mIoU), and to mix source and target data in the same batch in $\mathcal{L}^{\mathrm{sup}}$ ($-0.8$ mIoU), which encourages domain mixing (Chen et al., 2021a) and helps learn a more domain-robust segmentor.

Pixel contrastive learning is also found to be a good regularizer, removing $\mathcal{L}^{\mathrm{PC}}$ results in $-1.2$ mIoU (Tab. 5). Furthermore, we try two variants of pixel contrastive learning. Firstly, in "$\mathcal{L}^{\mathrm{PC}}$: $+\mathcal{D}_u$" we adopt the contrastive learning module proposed by Alonso et al. (2021), which uses both labeled and unlabeled data, but observe a performance drop ($-0.5$ mIoU). We attribute the drop to incorrect contrastive pairs on unlabeled pixels, while supervised pixel contrast only relies always on ground-truth. In the second variant, "$\mathcal{L}^{\mathrm{PC}}$: $+\mathcal{D}_s$", we try adding source labeled data to pixel contrast, without success ($-0.5$ mIoU). Some previous SSDA works even discourage source clusterability (Qin et al., 2021), aiming for source clusters to enclose target representations.

Finally, we report the improvement in performance between the model after the initial round of training ($\mathbf{M}_0$) and the final ensemble model after iterative offline self-training ($\mathbf{M}_1 + \mathbf{M}_2$), which brings +2 mIoU.

Table 5: Ablation study of the proposed framework on SSDA GTA→Cityscapes with 100 target labels (1/30). $\Delta$ denotes difference in mIoU to the baseline $\mathbf{M}_0$. Experiments are on the initial round of training (i.e., without iterative self-training). We note that consistency regularization is, by far, the most important component. All results are the average of 3 runs on a DeepLabv2 + ResNet-101 architecture.

| $\Delta$ | mIoU | Configuration | Steps |
|---|---|---|---|
| $-8$ | 56.0 | No $\mathcal{L}^{\mathrm{CR}}$ | 40k |
| $-2.3$ | 61.7 | $\mathcal{L}^{\mathrm{sup}}$: No class weight | 40k |
| $-1.2$ | 62.8 | No $\mathcal{L}^{\mathrm{PC}}$ | 40k |
| $-0.8$ | 63.2 | $\mathcal{L}^{\mathrm{sup}}$: No batch mix | 40k |
| $-0.5$ | 63.5 | $\mathcal{L}^{\mathrm{PC}}$: $+\mathcal{D}_u$ (Alonso et al. 2021) | 40k |
| $-0.5$ | 63.5 | $\mathcal{L}^{\mathrm{PC}}$: $+\mathcal{D}_s$ | 40k |
| $0$ | 64.0 | $\mathbf{M}_0$ | 40k |
| $+2$ | 66.0 | $\mathbf{M}_1 + \mathbf{M}_2$ | 120k |

Table 6: Impact of iterative self-training (rounds of 40k steps) vs. training for longer (one round of 120k steps) on SSDA GTA→Cityscapes with 50 target labels. Results are the average over 3 runs on a DeepLabv2 + ResNet-101 network.

| Steps | Model (Self-training) | mIoU | Model (longer training) | mIoU |
|---|---|---|---|---|
| 40k | $\mathbf{M}_0$ | 61.4 | $\mathbf{M}_0{}^*$ | 60.9 |
| 80k | $\mathbf{M}_1$ | 63.7 | $\mathbf{M}_0{}^*$ | 62.9 |
| 120k | $\mathbf{M}_2$ | 63.9 | $\mathbf{M}_0{}^*$ | 63.5 |
| 120k | $\mathbf{M}_1 + \mathbf{M}_2$ | **64.3** | - | - |

$^*$ Learning rate decayed linearly during 120k steps

### 4.3.2 Iterative self-training

In Fig. 4 we break down the impact of self-training. The first round of self-training (between $\mathbf{M}_0$ and $\mathbf{M}_1$) is the most effective, while the second round offers marginal to no improvement, indicating convergence of the self-training algorithm. Finally, the ensemble of $\mathbf{M}_1$ and $\mathbf{M}_2$ yields the best final performance.

We hypothesize that the main benefit of using pseudolabels is increasing diversity in target samples, which becomes more valuable at low labeling ratios, explaining the larger benefit at 50 target labels. It is also positive to drop pseudolabels after $n_{\mathrm{drop}}$ steps, compared to its counterpart (indicated in Fig. 4 as "No PL drop"), and fine-tune on ground-truth annotations.

In Tab. 6 we compare the self-training scheme to a single longer training round of 120k steps, for the case of 50 target labels. We find that a self-training scheme was both more effective, as it offered a better final performance (+0.8 mIoU), and more efficient, since at 80k steps it already outperformed a single 120k steps training round. The different learning rate decay schedules explains the difference between $\mathbf{M}_0$ at 40k steps, the more aggressive decay in self-training allows the model to fine-tune before.

### 4.3.3 Source styling and consistency regularization

Finally, we explored source styling to improve domain alignment. Source styling consists on transforming source images to adopt the target domain style, thus reducing the domain gap in the input space. We tried

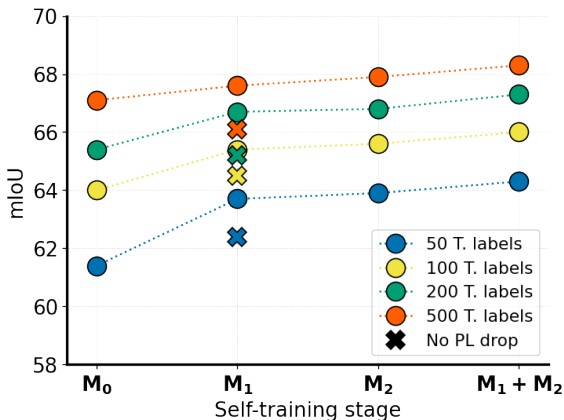

Figure 4: Evolution of performance during self-training from Algorithm 1. The first self-training round ($\mathbf{M}_0 \rightarrow \mathbf{M}_1$) brings the largest improvement, the final ensemble ($\mathbf{M}_1 + \mathbf{M}_2$) provides the best performance, and dropping pseudolabels for fine-tuning is beneficial. Results are an average over 3 runs for GTA→Cityscapes on a DeepLabv2 + ResNet-101 network. A tabular version can be found in Tab. 9.

Table 7: Study of the interaction between source styling and consistency regularization (CR). We observe how source styling is beneficial without CR, but harmful when combined with CR, as we hypothesize CR already brings robustness to style. Results are mIoU on the initial round of training $\mathbf{M}_0$, an average of 3 runs for experiments on SSDA GTA→Cityscapes using 100 target labels.

| $\Delta$ | mIoU | $\mathcal{L}^{CR}$ | Source styling |
|---|---|---|---|
| 0 | 56.0 | No | No |
| +1.4 | 57.4 | No | LAB (He et al., 2021) |
| +0.4 | 56.4 | No | photorealistic (Richter et al., 2022) |
| 0 | 64.0 | Yes | No |
| −0.3 | 63.7 | Yes | LAB (He et al., 2021) |
| −1.2 | 62.8 | Yes | photorealistic (Richter et al., 2022) |

two transformations, an online normalization of the LAB colorspace (He et al., 2021) (details in Sec. H) and replacing the original GTA images to GTA stylized as Cityscapes via photorealistic enhancement (Richter et al., 2022). In Tab. 7 we study the interaction of source styling with consistency regularization. When $\mathcal{L}^{CR}$ is not used, source styling helps, with LAB being most effective. Interestingly, source styling did not help when combined with $\mathcal{L}^{CR}$. We hypothesize that, since consistency regularization encourages similar predictions between images under strong augmentations, it may already be promoting a style-invariant model, to a point where styling source data is redundant. On the other hand, artifacts introduced by styling could be harming performance. This observation suggests that consistency regularization is not only promoting compact clustering but also encouraging domain robustness.

## 5 Discussion

In this paper, we revisit the SSDA setting in semantic segmentation, which has significant practical implications for industrial and medical imaging applications. We propose a simple SSDA framework that effectively uses the different kinds of data available and achieves fully-supervised accuracy using only a fraction of the target labels. Our method outperforms all SSDA baselines and demonstrates the high value of a handful of target labels to close the gap to supervised performance at a low annotation cost. Our results also demonstrate the generalization ability of the method to other datasets, even without further hyperparameter tuning. In addition, we provide insights into several important questions for segmentation practitioners and researchers who aim to minimize annotation costs. These include results on the scalability of existing UDA methods to the semi-supervised setting, as well as a comparison of SSDA and SSL in both low- and high-label regimes. Furthermore, in the following paragraphs, we discuss the relation of SSDA to both UDA and SSL, and propose ways to possibly adapt existing methods to SSDA.

We have demonstrated that existing UDA methods do not perform optimally in the semi-supervised regime, requiring methods tailored to SSDA. To adapt UDA frameworks to SSDA, we propose to consider an objective that emphasizes the tight clustering of target representations, which can be achieved through regularization with supervised pixel contrastive learning. Our findings suggest that domain alignment is less important in SSDA than achieving compact clusters of representations and then identifying them from few-shot samples, as also found by Mishra et al. (2021) in image classification. Having demonstrated the potential of SSDA, we encourage future DA research, mostly focused on UDA, to explore SSDA extensions and report results for varying numbers of target labels, in an effort towards a unified learning framework for unlabeled data, similar to Berthelot et al. (2021) for image classification.

Lastly, we observed that SSDA outperforms SSL in the low-label regime, but its advantage diminishes as the number of target labels increases. Practitioners facing a performance-vs-cost trade-off may be guided by Fig. 3 to choose between compiling a source-domain dataset (SSDA) or assuming a larger annotation cost and using SSL. Our experiments reveal that to effectively leverage a source dataset, an SSL method must account for domain alignment. In Tab. 5, we demonstrate that mixing domains in the supervised batch and using exclusively supervised pixel contrast can enhance SSDA performance.

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
