# OpenReview forum: "The Last Mile to Supervised Performance: Semi-Supervised Domain Adaptation for Semantic Segmentation"
_TMLR — Rejected by TMLR_

### Review · Reviewer_tMQ5 · 2023-10-10

**Summary Of Contributions:**

The paper proposes a Semi-Supervised Domain Adaptation (SSDA) training framework for semantic segmentation.  The paper's main contribution is to use a combination of training techniques: consistency regularization, pixel contrastive learning, and self-training to achieve new SOTA performance on SSDA semantic segmentation. Specifically, consistency regularization encourages features of the same image from different augmentation to cluster tightly. The pixel contrastive loss encourages positive pairs of labeled target data to cluster closer and negative pairs to spread out. Finally, the paper uses a second stage of self-training to use pseudo-labels to improve performance further.

**Audience:**

Yes

**Broader Impact Concerns:**

The work does not have obvious ethical implications.

**Claims And Evidence:**

Yes

**Requested Changes:**

* The authors should add more discussion to existing works, especially [Chen et al. 2021].  For example, why domain mixing is not as good? Would domain mixing help improve the performance of the proposed framework as well?


* Overall, the current writing gives the impression of simply applying existing techniques to a less explored area. Even though the paper demonstrated promising performance, which is a valid contribution, the paper needs to strengthen the unique challenges and aspects of SSDA for segmentation in terms of writing and experiments. For example, the related works section just plainly introduced works from UDA and SSL without discussing their strengths and weaknesses. It's not clear if they are applicable to SSDA for segmentation and why not applicable.

**Strengths And Weaknesses:**

**Notation**: in the following review, following the notations in the paper, $(\mathcal{D}_s,\mathcal{D}_t,\mathcal{D}_u)$ denote the source labeled data, target labeled data and target unlabeled data.

**Strength**
* The paper is clearly written and easy to follow.
* The various components are properly ablated.
The method introduced three losses: a supervised loss on $\mathcal{D}_s,\mathcal{D}_t$, a consistency loss on $\mathcal{D}_u$, and a contrastive loss $\mathcal{D}_t$. The effectiveness of the supervised loss on $\mathcal{D}_s,\mathcal{D}_t$ is demonstrated in Sec 4.2.4, where the proposed method performs much better with source-labeled data ($\mathcal{D}_s$) than without them.  The effectiveness of the other two losses is ablated in Sec 4.3.1. The paper found that consistency loss is the most effective component. The contrastive loss provides some improvement only when the labeled target data ($\mathcal{D}_t$) is used. Finally, the paper also clearly presents the improvement from the second stage of self-training using pseudo-labels in Sec.4.3.2.

* Empirically, results seem promising. On the GTA $\rightarrow$ Cityscapes benchmark,  it outperforms prior work on SSDA by a large margin and beats a competitive baseline extended from UDA.

**Weakness**

* The paper adopts several well-known techniques from semi-supervised learning (SSL) and unsupervised domain adaptation (UDA) to SSDA for semantic segmentation. However, the related works and discussions do not present specific challenges in this setting compared to SSDA for image classification, which has been explored more widely.

* This brings the question of what makes SSDA for semantic segmentation more challenging, do those unique challenges prevent us from using established SSDA pipelines [1,2,3,4], and how this paper addresses those challenges *differently* from existing pipelines. For example, consistency regularization and self-training bring the most improvement in the segmentation. These techniques are already widely used in UDA and SSL research and have been shown to improve performance. When using these methods in SSDA for segmentation, are there *unique* constraints and observations when compared to simple classification?

[1] Berthelot, David, et al. "Adamatch: A unified approach to semi-supervised learning and domain adaptation." arXiv preprint arXiv:2106.04732 (2021).

[2] Saito, Kuniaki, et al. "Semi-supervised domain adaptation via minimax entropy." Proceedings of the IEEE/CVF international conference on computer vision. 2019.

[3] He, Gewen, et al. "Classification-aware semi-supervised domain adaptation." Proceedings of the IEEE/CVF Conference on Computer Vision and Pattern Recognition Workshops. 2020.

[4] Kim, Taekyung, and Changick Kim. "Attract, perturb, and explore: Learning a feature alignment network for semi-supervised domain adaptation." ECCV 2020.

---

### Review · Reviewer_j5PW · 2023-11-02

**Summary Of Contributions:**

This paper proposes a semi-supervised domain adaptation framework for semantic segmentation. The framework consists of supervised training with labels, contrastive learning, and iterative self-training. The proposed method outperforms the existing SSDA methods on the  GTA→Cityscapes benchmark dataset, and fully supervised model on several other syn2real benchmarks.

**Audience:**

Yes

**Broader Impact Concerns:**

No ethical concerns.

**Claims And Evidence:**

Yes

**Requested Changes:**

Please consider revising the paper to address the weaknesses, particularly in:

- Clarifying the training process with more details.

- Deep diving the comparison between DAFormer and the proposed method.

- Analyzing the process of self-training and providing more details on pseudo labels and the progression of model performance.

**Strengths And Weaknesses:**

Strengths:

- The paper explores a less-studied area of semi-supervised domain adaptation (SSDA).

- The authors conduct comprehensive experiments on multiple benchmark datasets, demonstrating the effectiveness of their SSDA framework. The inclusion of an ablation study further adds to the paper's strength, as it sheds light on the impact of each module within the framework.

Weaknesses:

- Lack of detail regarding the minimization of the loss function in equation 7. The paper doesn't provide sufficient information about how different loss terms, originating from different datasets, are sampled during training. Furthermore, it's unclear why CR is applied only to D_u and not D_t. These aspects need further elaboration.

- The proposed method performs worse than DAFormer in some settings. Understandably, DAFormer works better in the UDA case, but its superior performance when the number of labels grows undermines the capability of the proposed method.  Additionally, the supervised training method shown in Figure 1 needs a clearer description. Why is it worse than the proposed method under 500 labels? Is it a fair comparison?

- The quality of pseudo-labels is a critical aspect of self-training, as the effectiveness of self-training heavily depends on the reliability of pseudo-labels. A discussion on the quality control of pseudo-labels should be included.

- In Algorithm 1, there is k = 1, ... K rounds. However, the paper only reports the experiments on M1 and M2. Does the performance converge after 2 rounds? Will training for more rounds lead to better results?

- Missing some references on previous papers using self-training in semantic segmentation.

Liu, Sheng, et al. “Adaptive Early-Learning Correction for Segmentation from Noisy Annotations.” CVPR 2022

---

### Review · Reviewer_B86a · 2023-11-19

**Summary Of Contributions:**

The paper presents a simple SSDA framework for semantic segmentation that effectively utilizes different types of available data. It demonstrates a significant improvement over UDA and discusses the suboptimal nature of existing UDA/SSL methods in the SSDA setting.  I am borderline on this paper. The results are good but the novelty is limited. The proposed approach is incremental.

**Audience:**

Yes

**Broader Impact Concerns:**

The paper does not seem to have much concerns on broader impact.

**Claims And Evidence:**

Yes

**Requested Changes:**

- The negative impact on the UDA setting should be addressed and mitigated. It would be helpful to explore potential solutions or adjustments that can improve the performance of the proposed approach in this context.
- Given the challenges associated with combining multiple losses, it is recommended to thoroughly investigate and fine-tune the hyperparameters to obtain the best possible performance. Providing insights or guidelines on how to effectively tune these hyperparameters can lead to better results.
- Exploring the impact of different hyperparameters on the final performance would be valuable in understanding the mechanism behind the approach and further refining its effectiveness.

**Strengths And Weaknesses:**

## Strength
- Present a simple SSDA method and shows its promise.
- Demonstrates state-of-the-art performance in using SSDA for semantic segmentation with minimal annotation in the evaluation on GTA &Cityscapes.
- The paper also explores the relationship between SSL and SSDA, showing improved performance when leveraging source domain data.

## Weakness
- Considering the existence of SSDA as a previously introduced approach, it would be beneficial to clearly highlight the unique aspects and contributions of the proposed method to enhance its novelty.
- I think the core issue here is how realistic is the setting? How this method contribute to practical problems.

---

> ### Author Response · Authors · 2023-11-23
> **Response to Reviewer B86a**
>
> Dear Reviewer,
>
> Thank you for providing feedback for our paper. Below we comment on the weaknesses and requested changes:
>
> **Weakness 1**: *Considering the existence of SSDA as a previously introduced approach, it would be beneficial to clearly highlight the unique aspects and contributions of the proposed method to enhance its novelty.*
>
> Our work presents a simple yet effective framework for SSDA. We agree with the reviewer that the setting is not new. However, we do make the point that the setting has a high potential and is mostly overlooked by the community. While build on a number of existing mechanisms, the framework is new to our knowledge. We support the design of our method with comprehensive ablation studies to show the impact of each component.
>
> Lastly, we would like to mention that as per TMLR’s Acceptance criteria, *'novelty of the studied method is not a necessary criteria for acceptance. We explicitly avoid these terms (“significant”, “impactful”, “novel”), and focus instead on the notion of “interest”'*. We believe that proposing an effective SSDA method is of undeniable interest for practitioners.
>
> **Weakness 2**: *I think the core issue here is how realistic is the setting? How does this method contribute to practical problems?*
>
> In our opinion, this setting is just as relevant for practitioners, if not more, than UDA. In most of the production applications, models have to be validated and cannot be used without evaluation on *target* data (e.g., in medical applications, we cannot use models to make critical decisions without a measure of how accurate the model is). Therefore, it is often the case that practitioners are already collecting *target labeled data* for evaluation purposes. Increasing the effort and annotating an additional fraction of data for training purposes (i.e., moving from UDA to SSDA) is, therefore, reasonable. Our work can serve as a reference for practitioners to compare UDA vs SSDA, and estimate if the benefit in performance of SSDA compensates the extra annotation effort. For the example of GTA2Cityscapes benchmark, where annotating 50 target samples brings +6.9 mIou points compared to the UDA state-of-the-art, it arguably does. Moreover, we evaluate on other DA benchmarks (without performing additional hyperparameter tuning) and show good performance, demonstrating off-the-shelf practical value in new datasets.
>
> **Requested Change 1**: *The negative impact on the UDA setting should be addressed and mitigated. It would be helpful to explore potential solutions or adjustments that can improve the performance of the proposed approach in this context.*
>
> We agree with the reviewer that a unified method for SSDA and UDA would be of great interest. Nevertheless, a unifying framework that considers the rich literature of UDA along with the trade-off of minimizing the annotation cost of SSDA is beyond the scope of our work. Our main target here is to simply motivate the community and practitioners to consider the SSDA setting more. Still, even if our method is not designed for UDA, notice that our method achieves a competitive performance, comparable to influential UDA methods such as Tranheden et al., 2021.
>
> **Requested Changes 2 and 3**: *Given the challenges associated with combining multiple losses, it is recommended to thoroughly investigate and fine-tune the hyperparameters to obtain the best possible performance. Providing insights or guidelines on how to effectively tune these hyperparameters can lead to better results. Exploring the impact of different hyperparameters on the final performance would be valuable in understanding the mechanism behind the approach and further refining its effectiveness.*
>
> Thanks for the remark. It is true that tuning the weights of the loss terms is important and can have a huge impact on the training. Other than the standard hyperparameters of deep learning algorithms, in our method we introduce two important parameters, the weight for the consistency regularization loss ($\lambda_1$) and the weight for pixel contrastive learning ($\lambda_2$). These parameters will depend highly on the application, so we can only provide a default which worked well in our case ($\lambda_1=1$, $\lambda_2=0.2$). When tuning them, we found a good indicator to look at the magnitude of the loss for each loss term, such that all terms are approximately balanced. It is also interesting to note that we tuned hyperparameters for the GTA2Cityspaces benchmark, and then used them on the rest of datasets without further tuning and with good performance, suggesting that these might be good default values. We have updated the implementation details in the appendix to include a discussion on the importance of these hyperparameters and guidelines to tune them. Please let us know if you would like us to conduct an additional experiment on this front.
>
> We are thankful to the reviewer and we would be glad to address any remaining concerns of the reviewer.

---

### Decision · Action_Editor_1jGT · 2024-01-05

**Recommendation:** Reject

**Comment:**

The most contentious part of the review was how this submission differs from existing literature, especially in terms of the technical approach. Multiple reviewers noted that this work proposes to use a combination of well-established techniques rather than developing a new technique that has not been studied before -- this by itself shouldn't be a reason for rejection, though it should be strongly justified for its significance (does this paper bring substantially new insights about existing techniques?). One reviewer pointed out that it is unclear what makes SSDA for semantic segmentation different from SSDA for image classification (where it is more well established than semantic segmentation), and what specific challenges this work addresses compared to existing SSDA approaches (and where & why they are lacking). Another reviewer made similar comments, criticizing that it is unclear what the unique aspects and contributions are in this work.

The authors provided a response, but the reviewers did not find it strongly convincing enough to change their recommendation. As a result, two reviewers recommended (weak) rejection, while one recommended (weak) acceptance. Unfortunately, none of the reviewers was willing to champion the submission.

**Audience:**

The reviewers noted that the studied problem of semi-supervised domain adaptation (SSDA) is under-explored. This submission could be of interest to some individuals in TMLR's audience who work on semantic segmentation in the SSDA setting.

**Claims And Evidence:**

The main claim, i.e., that one can improve semantic segmentation in the semi-supervised domain adaptation (SSDA) setting using a combination of existing training techniques (consistency regularization, pixel contrastive learning, and self-training), is well supported by good empirical performance on established benchmarks.

However, there were several concerns regarding the clarity. The most contentious point was on how this submission differs from prior art, especially in terms of technical challenges it addresses and where/why prior approaches are lacking. There was also a concern about the lack of empirical evidence supporting the efficacy of self-training with respect to the quality of pseudo labels.